# The Role of BRAF in Metastatic Colorectal Carcinoma–Past, Present, and Future

**DOI:** 10.3390/ijms21239001

**Published:** 2020-11-26

**Authors:** Angela Djanani, Silvia Eller, Dietmar Öfner, Jakob Troppmair, Manuel Maglione

**Affiliations:** 1Clinical Division of Gastroenterology, Hepatology and Metabolism, Department of Internal Medicine, Medical University Innsbruck, 6020 Innsbruck, Austria; angela.djanani@i-med.ac.at; 2Department of Visceral, Transplant and Thoracic Surgery, Medical University Innsbruck, 6020 Innsbruck, Austria; silvia.eller@i-med.ac.at (S.E.); dietmar.oefner@i-med.ac.at (D.Ö.)

**Keywords:** colorectal cancer, multitargeted therapy, BRAF inhibitors, BRAF, MAPK

## Abstract

With a global incidence of 1.8 million cases, colorectal cancer represents one of the most common cancers worldwide. Despite impressive improvements in treatment efficacy through cytotoxic and biological agents, the cancer-related death burden of metastatic colorectal cancer (mCRC) is still high. mCRC is not a genetically homogenous disease and various mutations influence disease development. Up to 12% of mCRC patients harbor mutations of the signal transduction molecule BRAF, the most prominent being BRAF^V600E^. In mCRC, BRAF^V600E^ mutation is a well-known negative prognostic factor, and is associated with a dismal prognosis. The currently approved treatments for BRAF-mutated mCRC patients are of little impact, and there is no treatment option superior to others. However, the gradual molecular understanding over the last decades of the extracellular signal-regulated kinase/mitogen-activated protein kinase pathway, resulted in the development of new therapeutic strategies targeting the involved molecules. Recently published and ongoing studies administering a combination of different inhibitors (e.g., BRAF, MEK, and EGFR) showed promising results and represent the new standard of care. In this review, we present, both, the molecular and clinical aspects of BRAF-mutated mCRC patients, and provide an update on the current and future treatment approaches that might direct the therapy of mCRC in a new era.

## 1. Introduction

Colorectal cancer (CRC) is still one of the leading cancers worldwide. With a global incidence of approximately 1.8 million cases and 700,000 cancer-related deaths per year, it is the third most prevalent form of cancer and the fourth most frequent cause of cancer-related death, only exceeded by lung, liver, and stomach cancers. By gender, CRC is the second most common cancer in women (9.2%) and the third in men (10%) [1]. Most cases of CRC are detected in Western countries (55%), but this tendency is changing due to the fast development of some countries over the past few years [2].

The lifetime risk to develop CRC is about 4% to 5% [3]. Alongside many personal traits or habits that are considered to be risk factors for developing polyps and in further sequence, CRC, the main risk factor remains age—past the fifth decade of life, the risk of developing CRC is markedly increased, while the onset of CRC below the age of fifty is rare (apart from inherited cancers) [4]. However, in recent years, the incidence in this age group increased, while there seems to be a slow decrease in the population above 50 years of age. Broader participation in screening programs is presumably the reason for these dynamics [5]. Other important risk factors are a history of inflammatory bowel disease or the presence of a positive familial history of CRC. Increased risk due to familial history can be derived from inherited mutations or the environment [6]. 

Most CRC patients with metastatic disease are treated with a combination of cytotoxic and biological agents. First-line chemotherapy with palliative purposes comprises fluoropyrimidines (e.g., 5-fluorouracil (5-FU) or capecitabine) alone, or combined with leucovorin (LV), as well as other cytotoxic agents, such as oxaliplatin (5-FU/LV/oxaliplatin (FOLFOX) and capecitabine/LV/oxaliplatin (CAPOX)), or irinotecan (5-FU/LV/irinotecan (FOLFIRI: FOLFIRINOX) [7,8,9,10,11].

After progression, patients with a good organ function and performance status (Eastern Cooperative Oncology Group ECOG 0-1) are offered a second-line chemotherapy regime, depending on the first line. Second-line treatment for patients refractory to irinotecan consist of an oxaliplatin-containing combination, whereas patients refractory to oxaliplatin are treated with an irinotecan-containing treatment [11]. The treatment option after triplet-therapy is not clearly defined. Alternatives consists of treatment with regorafenib [12] or trifluridine/tipiracil [13].

In addition to chemotherapy, monoclonal antibodies or proteins against vascular endothelial growth factor (VEGF) [14,15,16] and epidermal growth receptor (EGFR) [17,18], combined with traditional chemotherapy were demonstrated to improve the outcome of mCRC. 

Several drugs and combinations thereof are now available for the treatment of patients with advanced CRC, however, the optimal sequence of therapy remains to be established.

For the sake of completeness, local treatments should be mentioned as well, since they are an integral part of the multimodal concepts that could be offered to mCRC patients. Recent studies highlight their importance with either laparoscopic or open resection of liver metastasis, as well as percutaneous radiofrequency ablation, improving the survival rates in these patients. [19,20,21,22] 

However, mCRC is not a genetically homogenous disease and various mutations influence disease development, treatment response, and outcome. A prominent molecular feature is the BRAF mutational status. BRAF mutations occur in 8% of all tumors, and 5–12% of the mCRC patients present with a BRAF mutation [23]. More than 90% of them harbor the BRAF^V600E^ mutation associated with resistance to standard treatment regimens, and with a dismal prognosis [24]. In the light of the recently approved targeted therapies for the BRAF^V600E^-mutated mCRC, we present in this review, the molecular and clinical aspects related to this subgroup of patients.

## 2. BRAF—Molecular Insights and Clinical Relevance 

Colon cancer development results from the sequential accumulation of genetic alterations, which drive the progression from a benign stage (adenoma) to the fully transformed phenotype [25]. These genetic alterations underlie the manifestation of the hallmarks of cancer [26], which are essential for tumor initiation and progression. Mutations in intracellular signaling pathways, which when unperturbed are required for developmental processes and proliferation, survival, and differentiation of cells during postnatal life, function as essential drivers in colon carcinogenesis. Important entities affected include Wnt, RAS-RAF, PI3K/PKB/AKT, TGF-β, p53, and DNA mismatch-repair pathways [27].

### 2.1. Intracellular Signaling Pathways Involved in CRC

#### 2.1.1. RAS-RAF Pathway

BRAF is a member of the RAF kinase family, which additionally comprises ARAF and CRAF [28,29,30]. These serine/threonine kinases are part of an evolutionarily conserved pathway that connects the stimulation of cell surface receptors with intrinsic tyrosine kinase activity (Receptor Tyrosine Kinase, RTK, e.g., the epidermal growth factor receptor (EGFR, HER, cERBB)), with the stimulation of small G proteins of the RAS family, the activation of RAF kinases, and the downstream effectors MEK1/2 and their substrates ERK1/2. (Figure 1) Frequently, the net outcome is the transcriptional activation of genes involved in the proliferation, survival, or differentiation of cells. Signaling through this pathway plays a key role in the developmental processes but also during adult life, when components of this cascade can be affected by mutations in human cancers usually resulting in the constitutive activation of their enzymatic activity, which relieves them from control by extrinsic factors. 

Within the RAF family, BRAF is the preferred target for genetic alterations with the V600E exchange predominating. Mutation frequencies in human cancers are as high as 60% in malignant melanoma [30]. BRAF^V600E^ or BRAF^D594G^ exchanges that are mutually exclusive with the more frequent Kirsten rat sarcoma viral oncogene homolog (KRAS) mutations (30–50%) are present as oncogenic drivers in 5–12% of patients with mCRC [23]. Several studies demonstrated that the BRAF^V600E^ but no other less common BRAF mutations, are associated with a worse prognosis for these patients [31]. Stratification of CRCs based on gene expression resulted in the identification of four consensus molecular subtypes (CMS), with distinct features. CRCs carrying BRAF^V600E^ mutations are enriched in the subgroup CMS1 associated with hypermethylation, microsatellite instability (MSI), and chromosomal instability [32]. 

Mutations in BRAF are most commonly associated with an increase in its kinase activity, resulting in continuous downstream signaling. Therefore, several generations of mutation-specific small molecule BRAF inhibitors were developed, initially mainly for the use in the treatment of malignant melanoma, where BRAF mutations are most common. However, the clinical response, at best, was transient but a cure was never achieved [33,34]. One major obstacle was the fast development of drug resistance through various mechanisms, which usually left the ability of the drug to inhibit BRAF kinase activity intact, but bypassed its effect on downstream signaling. Drug unresponsiveness also went along with the activation of signaling proteins outside the RAS-RAF axis [35], which is discussed here briefly. For the clinical routine, this suggests the future use of treatment regimens, which combine the simultaneous inhibition of several signaling pathways and, in the future, also checkpoint inhibitors, as recently demonstrated for melanoma [36,37].

#### 2.1.2. PI3K-PKB Pathway

Apart from the RTK-RAS-RAF-MEK-ERK pathway, multiple other intracellular signaling cascades can become drivers for tumor development. In the context of CRC, these include PI3K [38] and Wnt [39,40] signaling. The lipid kinase phosphatidylinositol 3-kinase (PI3K), binds the small RAS G proteins through the same effector domain as the RAF kinases, and thus, RTK signaling might result in the concomitant activation of the RAF and PI3K. Membrane-derived lipids generated by PI3K are essential for the activation of a family of kinases called AKT1-3 or PKBα,β,γ, which fulfill important functions in proliferation survival and differentiation [41] (Figure 1). PI3K/PKB/AKT contribute to oncogenic signaling, following the loss or inactivation of the phosphatase PTEN, which normally would terminate PI3K signaling (tumor suppressor), the mutational activation or amplification of PI3K, or mutation of AKT/PKB [38,42].

#### 2.1.3. Wnt Pathway

The Wingless-related integration site (Wnt) pathway is another evolutionarily conserved signaling cascade frequently implicated in oncogenesis. Wnt is a family of lipoglycoprotein ligands, which bind to the frizzled (FZD) family receptors. One main downstream effect of receptor activation is the stabilization of the cytosolic protein β-catenin, which regulates the expression of many cancer-relevant proteins (Figure 1). The adenomatous polyposis coli (APC) protein is part of the destruction complex, which is required to maintain low β-catenin levels in the absence of Wnt signaling. APC is mutated in 90% of all CRC patients, and frequently cooperates with mutations in KRAS and BRAF. Although deregulation of Wnt signaling is associated with the subgroup CMS2 of CRCs [32], several findings support the potential of simultaneously targeting both pathways in CRC. Stimulation by Wnt was shown to activate signaling through RAF-MEK-ERK and to assist in the stabilization of RAS proteins, thereby enhancing downstream signaling [43]. Furthermore, WNT5A promotor methylation and BRAF^V600E^ mutation are associated in CRC patients [44].

### 2.2. Targeted Therapies for CRC

The molecular understanding of the underlying genetic landscape of CRC provided the rationale basis for novel therapeutic approaches. This, in particular, includes the clinical use of small molecule inhibitors of mutant BRAF (e.g., encorafenib [45]) and of MEK1,2 (e.g., binimetinib [46]), combined with the inhibition of PI3K [47], or the epidermal growth factor receptor (e.g., cetuximab, [48]). EGFRs (HERs/cERBBs) are prototypic RTK receptors, upstream of RAF and PI3K signaling. Overexpression and mutation contribute to tumor progression. Evidence of cERBB2/HER2 amplification and mutation in CRC suggests it to be a potential therapeutic target [49].

### 2.3. Clinical Relevance of Molecular Testing in CRC

Since many genetic subtypes of CRC are associated with specifically targeted treatment options, molecular testing has become clinical routine (Figure 2). Reports of the predictive value of various mutational status highlighted the clinical relevance of molecular testing in CRC patients, in the last decade. Half of the patients with advanced CRC harbor a KRAS or a neuroblastoma N-Ras (NRAS) tumor gene mutation. These mutations are negative predictive biomarkers with regards to the treatment response to the anti-EGFR monoclonal antibodies cetuximab or panitumumab [17,18,50,51]. Since RAS proteins belong to the main effectors of EGFR signaling, the presence of mutationally-activated RAS might bypass the effect of inhibiting EGFR signaling. Therefore, only patients with RAS wild-type mCRC should receive a therapy that includes anti-EGFR treatment.

For localized, non-metastatic CRC, there are currently no data supporting the analyses of other disease markers than the microsatellite instability/DNA mismatch repair (MSI/MMR) status. While MSI/MMR status determination is important to rule out hereditary non-polyposis CRC (HNPCC, Lynch syndrome) and to identify patients with a low risk of recurrence, B-RAF, and KRAS analysis seems to not add further information in the treatment decision-making process [53].

In contrast, for mCRC, the standard panel of molecular markers comprises the MSI/MMR status, RAS, HER-2, and BRAF [9,54]. Due to the high immunogenicity shown in MSI tumors, MSI/MMR status determination is important to identify patients who will benefit from immune checkpoint inhibitors [52]. As already mentioned, RAS mutations identify patients resistant to anti-EGFR therapies. Furthermore, recent data identified HER-2 amplification as a possible treatment target in mCRC patients, not responding to standard chemotherapy lines [55]. Last but not the least, BRAF mutations are the focus of recent and current clinical trials, where specific targeted approaches are tested in mutated mCRC.

More than 90% of mutations in BRAF-mutated cancers occur in codon 600 (V600E mutation). The so-called non-V600E-BRAF mutations in codon 594 and 596, account for less than 5% [56]. The reported incidence of BRAF^V600E^ mutation varies between 5% and 12% [57,58,59,60,61,62], even though recent registry data report 21% of mCRC patients harboring BRAF mutations [63]. The differences arise from differences in the tumor stages included in the reporting papers, with a stronger decline in the advanced tumor stages, due to their worse prognosis. Interestingly, RAS and BRAF mutations are mutually exclusive, and are reported to occur together in only 0.001% of patients [64]. 

Non-V600E-BRAF mutations define a clinically distinct subtype of CRC. These mutations occur more frequently in the left-sided colon and rectum, are associated less with peritoneal metastases, and were shown to be associated with a microsatellite stable (MSS) status. Even though they result generally in a significantly better overall survival (OS) (median 62.0 vs. 12.6 months; HR 0.36, *p* = 0.002) [65,66], non-V600E-BRAF-mutated CRC can be subdivided in two classes, with respect to their anti-EGFR treatment response, the *RAS*-*independent* activating (class 2) and the *RAS-dependent* activating non-V600E-BRAF mutation (class 3) [67].

In contrast, the BRAF^V600E^ mutation in colon cancer occurs more frequently in women and elderly patients, in proximal tumor locations, and in tumors arising from serrated adenomas and with mucinous differentiation. It is also associated with a higher rate of lymph node metastases and peritoneal dissemination [60,68,69,70]. From a molecular point of view, in up to 50%, it is associated with high microsatellite instability MSI-H [71]. Patients with BRAF^V600E^ mutation survive, on average, less than half as long as patients with BRAF wild-type mCRC. [59,60,72]

#### 2.3.1. Prognostic Value of BRAF^V600E^

BRAF^V600E^ mutation is known as a negative prognostic marker. 

Regarding non-metastatic CRC, the evaluation of more than 1300 specimens in the PETACC-3 trial, revealed BRAF^V600E^ mutation as marker for significantly worse OS (HR 1.78, 95% CI 1.15–2.76); however, it did not influence recurrence-free survival (RFS) (HR 1.30; 95% CI 0.87–1.95) [73]. Domingo et al. observed a shorter relapse-free survival for BRAF^V600E^ mutated patients (HR 2.21, 95% CI 1.47–3.29) [74], in a population combining the QUASAR 2 trial and an Australian community-based series. More recent data from the PETACC-8 and N0147 trials confirmed the negative prognostic value for both, time to recurrence (TTR) (HR 1.27, 95% CI 1.04–1.56) and OS (HR 1.49, 95% CI 1.20–1.86) [53].

The frequent occurrence of MSI in BRAF^V600E^ mutation, poses the question of whether the MSI status could act as a possible opposite prognostic factor in the BRAF^V600E^-mutated patients. Indeed, despite the small number of events, PETACC-3 trial data suggest that MSI-H status overrules the prognostic value of the BRAF^V600E^ mutation status (RFS: HR 1.26, 95% CI 0.59–2.70; OS: HR 1.53, 95% CI 0.63–3.70) [73,75]. The analysis of 1913 stage II specimens of the QUASAR trial showed that the BRAF^V600E^ mutation status did not influence the better RFS in the MSI/MMR tumors (HR 0.48, 95% CI 0.27–0.85) [76]. Similarly, recent data including PETACC-8 and the N0147 trial with 4411 patients confirmed BRAF^V600E^ mutation as a negative prognostic marker in stage III MMS patients (TTR: HR 1.54, 95% CI 1.23–1.92; OS: HR 2.01, 95% CI 1.56–2.57); however, with no prognostic influence on MSI patients (TTR: HR 0.94, 95% CI 0.58–1.51; OS: HR 1.26, 95% CI 0.78–2.04) [53]. Results from the intergroup trial CALGB 89803, reflect the difficult task of interpreting these data. Categorization according to BRAF, as well as MSI status, suggested opposing prognostic effects of BRAF^V600E^ mutation and MSI-H, however, no difference reached statistical significance [77]. In contrast, the analysis of stage III colon cancer patients of the N0147 trial did not support these findings [78]. 

The negative impact of BRAF^V600E^ mutation was also reported for patients with advanced CRC. A pooled analysis including more than 3000 patients of the CAIRO, CAIRO 2, COIN, and FOCUS trial, showed in patients with BRAF^V600E^ mutation, both worse progression-free survival (PFS) (HR 1.34, 95% CI 1.17–1.54) and OS (HR 1.91, 95% CI 1.66–2.19) [79]. Data from the AIO 0207 trial showed that the BRAF^V600E^ mutation remains a negative prognostic marker, with a significantly worse OS in right and left-sided colon cancer [80]. Similarly, data from the FIRE-3 study and the MRC FOCUS trial confirm a worse prognosis for PFS and OS in this patient group [59,81]. 

In contrast to stage II and stage III cancer, a recent pooled analysis including more than 3000 patients, suggests that the MMR status does not influence the prognostic value of the BRAF^V600E^ mutation in advanced CRC (advCRC) [79].

The prognostic value of the BRAF^V600E^ mutation is also reflected in the outcome of resectable colorectal liver metastases. A recent multicenter analysis reports a 93.9% recurrence rate, over a median follow-up period of almost 50 months, with an estimated 5-year OS rate of 18.2% [82]. Still, the observed long-term survivors highlight the necessity of a more granular stratification aimed at identifying patients suitable for specific local treatments. These stratifications should be based on the clinical markers [83], as well as on additional molecular-marker-like alterations in the SMAD family, as proposed by Lang et al. in their extended clinical score [84].

#### 2.3.2. Predictive Value of BRAF^V600E^

While KRAS mutation status is now widely accepted as a predictive marker for resistance towards anti-EGFR treatment [50], the predictive role of BRAF^V600E^ mutation towards chemoresistance is still under debate. Of note, the low prevalence of this mutation makes it difficult to establish it as a predictive marker.

Already earlier trials like the MRC FOCUS trial reported that the BRAF^V600E^ mutant tumors had a worse prognosis but no predictive value for PFS or OS, neither in the irinotecan/FU group nor in the oxaliplatin/FU group (*p* = 0.16 and *p* = 0.30), highlighting, however, that these results should not preclude those patients from intensified treatments [59]. Current guidelines of intensified chemotherapy regimens for patients bearing BRAF^V600E^ mutation, rely on the results of the phase III TRIBE study. In that study, the treatment with LV/5-FU/ oxaliplatin/ irinotecan (FOLFOXIRI) plus bevacizumab showed a significantly better OS and PFS, compared to FOLFIRI plus bevacizumab in the intention to treat (ITT) population, and a relevant clinical, despite not statistically significant, advantage in median OS (19.4 vs 10.7 months; HR 0.54, 95% CI 0.24–1.20). Again, the mutation had no predictive value (HR 1.89, 95% CI 0.38–8.78) [85]. 

The impact of anti-VEGF treatment in this subset of patients is not yet clear. Results from the phase III AGITG MAX trial showed that the BRAF^V600E^ mutation did not predict the effectiveness of Bevacizumab, if added to capecitabine (OS: *p* = 0.32 PFS: *p* = 0.46, for the interaction of BRAF status and the assigned treatment status) [86]. In the same line, the phase III study RAISE, failed to show any statistically significant predictive value of the BRAF mutation status, however, the OS and PFS doubled in patients treated with the VEGF receptor 2 antibody ramucirumab [87]. Similarly, the VELOUR trial showed a trend towards a significant increase of OS in the BRAF^V600E^-mutated patients treated with aflibercept. Of note, this difference was even more pronounced than in the RAS mutant and RAS wild-type subgroups, suggesting that BRAF^V600E^-mutated patients benefit from aflibercept [15].

Since BRAF is a downstream signaling protein of the EGFR-mediated mitogen-activated protein kinases (MAPK) pathway, the efficacy of anti-EFGR treatments was also challenged for the BRAF^V600E^ mutations. In contrast to the KRAS mutation status, the results were not that conclusive. The PRIME study showed that the addition of panitumumab to FOLFOX4 did not result in better PFS and OS for BRAF-mutated patients (HR 0.58, 95% CI 0.29–1.15; HR 0.90, 95% CI 0.46–1.76, respectively). Another important finding was, that the negative predictive value for PFS and OS observed in the patient group with either RAS or BRAF mutations was driven by the RAS-mutated patients [88,89]. The different way of interpreting the data was well reflected in two recent metanalyses. Despite both studies stating that BRAF^V600E^ mutation had no predictive value on median PFS or median OS, one study suggests mandatory BRAF mutation assessment before initiating anti-EGFR treatment [64], whereas the second study concludes that there is not enough evidence to support mandatory assessment [90]. The addition of panitumumab to irinotecan in a 2nd line treatment (PICCOLO trial) was suggested to even have a detrimental effect (PFS: HR 1.40, 95% CI 0.82–2.39; OS: HR 1.84, 95% CI 1.10–3.08). Due to the low case number, the authors define their results as exploratory [91]. In contrast, the VOLFI trial reports encouraging data, however, there were only 16 BRAF^V600E^-mutated patients included. The addition of panitumumab to modified FOLFOXIRI (mFOLFOXIRI), resulted in significantly higher overall response rate (ORR) as compared to mFOLFOXIRI alone (OR 21, 95% CI 1.5–293.2) [92]. Recent findings suggest that further stratifications by other predictive factors like tumor sidedness, might reveal patient subsets where the BRAF mutation status is predictive for treatment response to anti-EGFR treatment [93]. 

According to the current ESMO guidelines, the preferred choice for 1st line treatment in fit, BRAF^V600E^ mutant patients, is the triplet chemotherapy FOLFOXFIRI plus bevacizumab [9]. The German S3-guidelines also suggest an aggressive triplet treatment; however, they also point at the discordant results regarding the targeted anti-VEGF therapies and argue based on a recent subgroup analysis of the FIRE-3 trial that these patients might not benefit from either anti-EGFR- or anti-VEGF-based strategies [94]. The guidelines are now also challenged by the recently published results of the TRIBE2 trial. In this trial, the mentioned triplet cytotoxic regimen plus bevacizumab, did not show any significant benefit in the BRAF-mutated patients, as compared to the cytotoxic doublets in combination with bevacizumab [95]. 

With respect to the targeted treatments, both the German and the NCCN guidelines include the recent developments in the BRAF-targeted therapies [54,96]. 

## 3. Targeting BRAF in the mCRC Treatment

After the failure of 1st line of treatment, unfortunately, the subsequent lines only have a minimal effect on tumor development. Most of the time, patients experience rapid progressive disease (PD). There were many attempts to overcome further tumor progression, highlighting the unmet medical need for this group of patients [97,98].

Recently, BRAF inhibitors such as vemurafenib, dabrafenib, and encorafenib, revolutionized the treatment of BRAF^V600E^ metastatic melanoma, initially in monotherapy or, currently, in combination with other drugs. Ongoing studies are also striving to reproduce these results in patients with mCRC. 

### 3.1. Monotherapy–The Broken Promise

Unlike other tumors with BRAF^V600E^ mutations, like melanoma [99,100,101], non-small-cell lung cancer [102] and papillary thyroid cancer [103], BRAF inhibition in BRAF^V600E^ mutant mCRC showed only marginal clinical activity in the early treatment course. 

Kopetz et al. led one of the first trials with a BRAF inhibitor, in previously-treated BRAF-mutated mCRC, using the recommended phase II dose of *vemurafenib* for melanoma (960 mg b.i.d.) in an expansion cohort [45]. A total of 21 mCRC patients with confirmed BRAF^V600E^ mutations CRC were included. A confirmed partial response (PR) lasting 21 weeks and seven cases of stable disease lasting at least 8 weeks were reported. Median PFS was 2.1 months (range, 0.4–11.6months), with two patients being progression-free for more than 6 months. Median OS was 7.7 months (range, 1.4–13.1 months).

In a phase I basket trial of *dabrafenib*, 11 mCRC patients were included, however, only 9 had a BRAF-mutant evaluable disease. Of these, PR was observed in only 1 patient, while in 7 patients, there was a stable disease [104]. 

Similarly, treatment with *encorafenib*, which had a more prolonged pharmacodynamic activity than the other approved BRAF inhibitors, did not show encouraging results. In a phase I escalation study, none of the included 18 patients achieved PR or a complete response (CR). After a median treatment duration of 11 weeks, 14 patients had to discontinue the treatment, most of them due to PD. Median PFS was 4.0 months [105].

The lack of clinical effectiveness of BRAF inhibitor monotherapy is currently explained by two observations. In vitro studies suggest that BRAF inhibition causes a rapid feedback activation of EGFR because of the missing negative feedback mechanism driven by ERK1/2 activation, and, in contrast to melanomas, CRC express higher EGFR levels [106,107]. As a consequence, EGFR activates MEK1/2 through several escape mechanisms, e.g., bypassing BRAF via other RAF family members or via activation of the PI3K/AKT pathway, finally resulting in missing the tumor response [108] (Figure 3).

### 3.2. Multitarget Approaches to Overcome Resistance in BRAF Mutated mCRC

To overcome the limited activity in the BRAF^V600E^-mutated mCRC, different approaches were tested in several studies that simultaneously target various signaling entities, combining BRAF inhibitors, e.g., with anti-EGFR monoclonal antibodies and MEK inhibitors (Figure 4).

#### 3.2.1. Targeting BRAF and MEK

As it is well known that a combination of BRAF and MEK inhibition proved to be more effective in melanoma than only BRAF inhibition, the same approach was evaluated in the BRAF-mutated mCRC.

Corcoran et al. analyzed the combination of the selective BRAF inhibitor *dabrafenib* with *trametinib*, a selective MEK inhibitor, in patients with histologically confirmed BRAF^V600E^ or BRAF^V600K^-mutant mCRC, in a phase I/II trial [109]. A total of 43 patients were treated with dabrafenib (150 mg twice daily) plus trametinib (2 mg daily), 17 of whom were enrolled onto a pharmacodynamic cohort, undergoing mandatory biopsies, before and during treatment.

Five patients (12%) achieved a PR or better, including one (2%) CR, with a duration of response >36 months; 24 patients (56%) achieved stable disease as the best confirmed response. With a median PFS of 3.5 months, the efficacy was greater than the median PFS of 2.5 months, observed with standard chemotherapy [110]. 

#### 3.2.2. Targeting BRAF and EGFR

In a recent open-label phase one study, 20 BRAF^V600E^ mutant patients were treated with the BRAF inhibitor *dabrafenib* and the anti-EGFR monoclonal antibody *panitumumab*. Two patients achieved a CR or PR, while 16 patients had stable disease, resulting in a tumor control of 90%. Again, the median PFS was 3.5 months (95% CI, 2.8–5.8) [111].

A similar response rate (RR) (13%) and similar PFS 3.2 months (95% CI, 1.6–5.3) could be achieved with the combination of the BRAF inhibitor *vemurafenib* with *panitumumab*, in a pilot study involving 15 patients; despite being well-tolerated, clinical activity of this treatment was modest [48].

These results suggest that EGFR-independent mechanisms might lead to MAPK reactivation in BRAF and EGFR-targeted treatment strategies.

The SWOG 1406 study analyzed the addition of *vemurafenib* to the anti-EGFR (*cetuximab*)/chemotherapy (irinotecan) combination. The ORR improved dramatically in the triple-therapy, compared to the dual-therapy (16% vs. 4%), however, even though PFS was significantly longer (4.4 months (95% CI: 3.6–5.7) vs. 2.0 months (95% CI: 1.8–2.1); *p* < 0.001), this combination showed only moderate clinical effectiveness [112].

#### 3.2.3. Targeting BRAF and EGFR and PI3K

Interesting results were observed in a dose-escalation trial, where BRAF-mutated mCRC patients were administered the BRAF inhibitor *encorafenib* and the anti-EGFR monoclonal antibody *cetuximab*, with (28 patients) or without the PI3Kα inhibitor *alpelisib* (26 patients) [113]. Both treatment regimens resulted in a similar clinical efficacy. In the dual-combination and in the triple-combination, the ORR was 19% (one CR, four PR), and 18% (5 PR), respectively. However, the median PFS was similar to the before-mentioned treatments, with 3.7 months (95% CI, 2.8–12) for the dual- and 4.2 months (95% CI, 4.1–5.4) for the triple-combination.

#### 3.2.4. Targeting BRAF and MEK and EGFR

Up to now, two published triple-combinations inhibited BRAF, MEK, and EGFR.

The first trial, an open-label phase I study, analyzed the efficacy of BRAF and EGFR inhibition with *dabrafenib* and *panitumumab* combined with the MEK inhibitor *trametinib*, in 91 BRAF^V600E^ mutant mCRC patients. Of note, 23 patients did not have any prior line of therapy. This treatment strategy resulted in 19 patients experiencing CR or PR, and 59 patients having a stable disease. The median PFS was 4.2 months (95% CI, 4.0–5.6) and the OS was 9.1 months (95% CI, 7.6–20.0 months, estimable but not mature). The triple-combination was characterized by a 70% grade 3/4 adverse events [111].

In October 2019, the group of Tabernero published the results of the BEACON trial, the largest clinical study in this patient population. The trial met all its endpoints and is now included in NCCN (National Comprehensive Cancer Network) guidelines, as a recommended treatment after failure of one or two prior lines [114].

In this open-label phase 3 trial, they enrolled 665 patients with BRAF^V600E^–mutated mCRC, who showed disease progression after one or two previous regimens. Patients were randomly assigned in a 1:1:1 ratio. The triplet-therapy group received the BRAF inhibitor *encorafenib*, the MEK inhibitor *binimetinib*, and *cetuximab*. The doublet-therapy group was treated with *encorafenib* and *cetuximab*, and the third group received either cetuximab and irinotecan, or cetuximab and FOLFIRI, according to the investigators’ choice (the control group). 

The median OS was 9.0 months in the triplet-therapy group and 5.4 months in the control group (hazard ratio [HR] for death, 0.52; 95% confidence interval [CI], 0.39 to 0.70; *p* < 0.001). The confirmed RR was 26% (95% CI, 18–35) in the triplet-therapy group and 2% (95% CI, 0–7) in the control group (*p* < 0.001). Of note, the RR in patients with only one prior line of therapy was 34% (95% CI, 23–47). The median PFS was 4.3 months (95% CI, 4.1–5.2). The dual-combination achieved similar results, with a median OS of 8.4 months (HR for death vs. control, 0.60; 95% CI, 0.45–0.79; *p* < 0.001). Adverse events of grade 3 or higher occurred in 58% of patients in the triplet-therapy group, in 50% in the doublet-therapy group, and in 61% in the control group. 

An updated analysis of the study confirmed the clinical efficacy of these treatment regimens. Doublet and triplet-therapy achieved a median OS of 9.3 months (95% CI, 8.0–11.3) and 9.3 months (95% CI, 8.2–10.8), respectively, compared to 5.9 months (95% CI, 5.1–7.1) in the control group, showing for the first time a significant survival benefit of the targeted therapies in the BRAF^V600E^ mutant mCRC patients, as compared to the standard chemotherapy options.

However, there are also some criticisms about the design of the trial. A portion of patients in the control arm might have never received an oxaliplatin-containing regimen (e.g., FOLFOX), but they merely received two irinotecan-containing regimens, consecutively. Another criticism is the missing report about the exact treatment regimens in the control arm, and about previously administered and following treatment lines. Additionally, the non-provided information about the time lag between the diagnosis of the metastatic disease and study enrollment might have affected the study results.

Finally, there is the crucial question of whether triplet was better than doublet. Even though the trial was not powered to compare the 2 regimens, there was only 0.6 months longer median OS in the triplet therapy group. However, due to the higher incidence of grade 3/4 toxicity, the European Medicines Association so far only approved the cetuximab/encorafenib combination.

## 4. Future Perspectives

The encouraging results of the BEACON trial highlight the importance of multitargeted approaches in this specific patient population, also demonstrating, however, that there are other escape mechanisms of the tumor, leading to a still poor prognosis in BRAF^V600E^ mutant mCRC patients. The observation of a better RR in patients with only one prior line, raises the question of the timing. In this regard, this chemotherapy-free combination is being explored frontline in the ANCHOR clinical trial (NCT03693170).

Intervening in the Wnt/ß-catenin signaling, represents another potential future treatment option. Wnt was shown to activate signaling through RAF-MEK-ERK targeting [43], e.g., the S100 calcium-binding protein A4 (S100A4) [115]. S100A4 is associated with metastasis formation and reduced OS in CRC [116]. The phase II NIKOLO trial (NCT02519582) will test the efficacy of the antihelminthic drug niclosamide in controlling the progression of mCRC, via reduced expression of S100A4 [117]

Another important aspect of the BRAF^V600E^ mutant mCRC patients is the already mentioned co-occurrence of deficient-MMR, in up to 50%. Considering the encouraging results with the immune checkpoint inhibitors involving programmed cell death-1 (PD-1) protein-like pembrolizumab and nivolumab in MSI CRC patients [118], immunotherapy is to be considered in the BRAF^V600E^ mutant mCRC patients as well. In this regard, a positive correlation between the BRAF^V600E^ mutation and programmed death ligand-1 (PD-L1) was also recently described [119]. In the recently published KEYNOTE-146 (pembrolizumab) open-label phase II study, 14 out of 124 MSI-H/dMMR, included CRC patients that harbored a BRAF mutation. In 6 of these patients, an overall response could be observed [120]. Similarly, in the CheckMate 142 trial, the use of nivolumab, with or without the anti-CTLA4-antibody ipilimumab, resulted in an ORR of up to 55% [121,122]. Table 1 summarizes current trials addressing multitargeted approaches in BRAF mutated advCRC and mCRC.

## 5. Conclusions

Patients with mCRC harboring BRAF^V600E^ mutations are still burdened with a dismal prognosis compared to patients without this mutation. Current treatment options in these patients have insufficient clinical efficacy. The advent of treatment regimens addressing molecular targets in the signaling pathway is going to improve upfront treatment options, if not replace those based on cytotoxic agents.

The emergence of resistant subclones or escape mechanisms harboring MAPK-activating alterations might be a major driver for treatment failure and clearly shows that there is still a long way to go. Future strategies aimed at sustaining clinical benefit by suppressing these resistance mechanisms should include a deeper understanding of the molecular pathways, as well as combined approaches, not only addressing the targets of various intracellular signaling pathways, but also other currently available molecular characteristics like MMR.

## Figures and Tables

**Figure 1 ijms-21-09001-f001:**
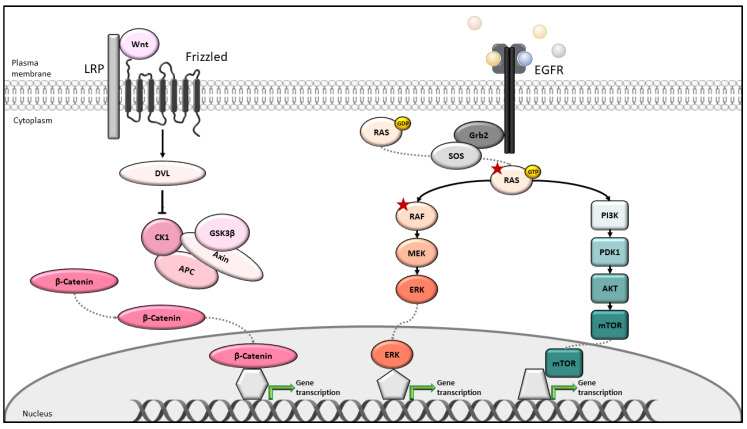
Schematic illustration of the canonical Wnt/β-catenin, RAS-ERK, and PI3K/AKT/mTOR signaling pathways. In the presence of extracellular Wnt ligands, the β-Catenin degradation complex is inhibited and β-Catenin translocates to the nucleus, resulting in the activation of the target genes. Additionally, Wnt can affect the RAF-MEK-ERK signaling through the stabilization of the RAS proteins. The RAS-ERK route is stimulated through the binding of EGF to EGFR, which then allows SOS to activate RAS by exchanging GDP to GTP. GTP-bound RAS is necessary for the activation of RAF and the signal is propagated to MEK-ERK kinase, via phosphorylation. Phosphorylated ERK translocates to the nucleus and activates various transcription factors. Activated PI3K, an additional RAS target, results in the activation of PDK1 and AKT. AKT signaling, in turn activates mTOR, leading to the expression of target genes. Red asterisks indicate the gain of the function mutation, ©Silvia Eller.

**Figure 2 ijms-21-09001-f002:**
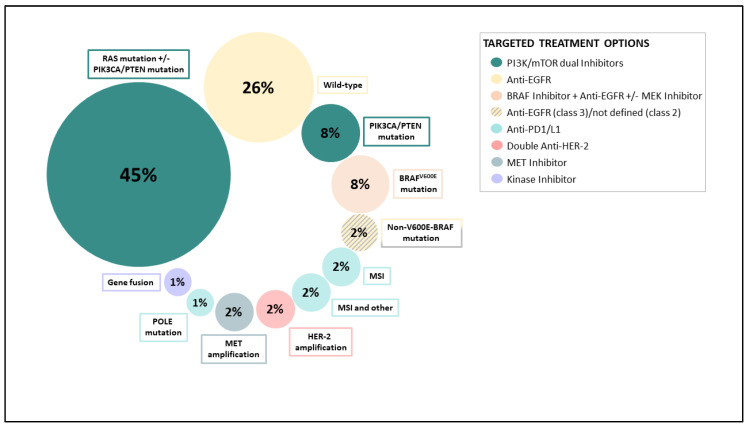
Most frequent genetic subtypes observed in mCRC and potential targeted treatment options (adapted from [52]), ©Silvia Eller.

**Figure 3 ijms-21-09001-f003:**
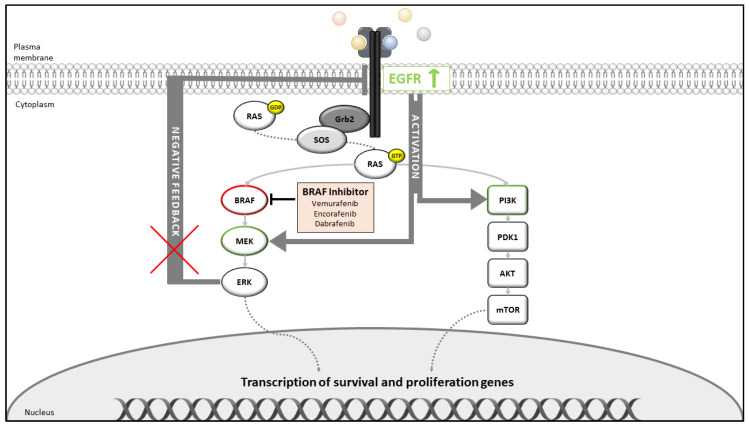
Feedback mechanisms of the BRAF inhibitor monotherapy. BRAF inhibitors suppress the ERK-mediated negative feedback phosphorylation of EGFR (x, red cross), leading to increased EGFR (↑, green arrow)/RAS activity, which then results in the activation of the RAF family member CRAF or the PI3K/AKT pathway, ©Silvia Eller.

**Figure 4 ijms-21-09001-f004:**
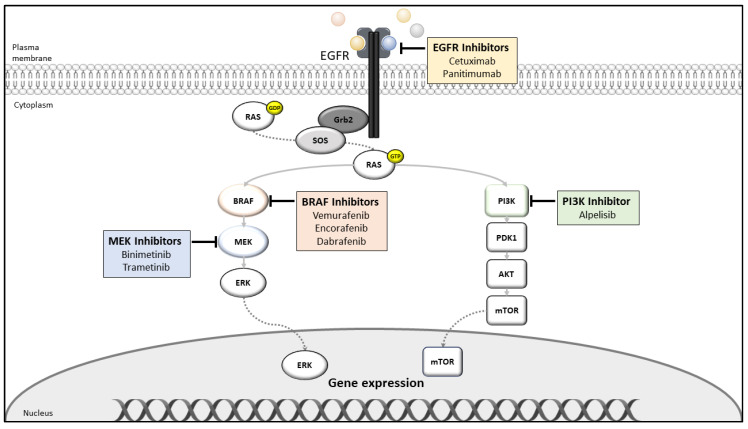
Current clinically applied targeted therapy options for mCRC, ©Silvia Eller.

**Table 1 ijms-21-09001-t001:** Summary of current ongoing trials including patients with BRAF mutation.

Targets	Compounds	Study Design	Phase	Inclusion Criteria	Participants	Primary Endpoints	Registration Number
BRAF + EGFR	vemurafenib + cetuximab + FOLFIRI	Open-label, single-arm	II	advCRC or recCRC	30	ORR	NCT03727763
BRAF + EGFR + MEK	encorafenib + cetuximab + binimetinib	Open-label, single-arm	II	first-line treatment in mCRC	95	ORR	NCT03693170
BRAF + MEK + PD-1	encorafenib + binimetinib + nivolumab	Open-label, single-arm	I/II	MSS mCRC, ≥1 treatment lines	38	(a) radiographic response (b) best investigator-assessed response (c) treatment-related grade ≥ 3 AEs	NCT04044430
BRAF + MEK + PD-1	dabrafenib + trametinib + PDR001	Open-label, single-arm	II	mCRC, ≥0 treatment lines	25	(a) ORR (b) treatment-related grade ≥ 3 AEs	NCT03668431
BRAF	oral LGX818	Open-label, single-arm	I	mCRC/mMelanoma	107	dose-limiting toxicities	NCT01436656
BRAF + EGFR + PI3K	(a) LGX818 + cetuximab (b) LGX818 + BYL719 + cetuximab	Open-label, parallel assignment	lb/II	mCRC, ≥1 treatment lines	156	(a) dose-limiting toxicities (b) PFS	NCT01719380
BRAF	oral ABM-1310	Open-label, sequential assignment	I	adv or met solid tumors including mCRC	27	(a) Maximum Tolerated Dose (b) Recommended Phase 2 Dose	NCT04190628
BRAF + EGFR + PD1	encorafenib + cetuximab + nivolumab	Open-label, single-arm	I/II	MSS mCRC, ≥1 ≤2 treatment lines	38	(a) ORR (b) treatment-related grade ≥ 3 AEs	NCT04017650
BRAF + MEK	LGX818 + MEK162	multicenter, open-label	Ib/II	adv or met melanoma, mCRC, ≥1 treatment lines	127	(a) dose-limiting toxicities (b) ORR	EudraCT Number: 2011-005875-17
EGFR or VEGF	(a) cetuximab + FOLFOXIRI (b) bevacizumab + FOLFOXIRI	Randomized	II	1st line mCRC	108	ORR	EudraCT Number: 2015-004849-11

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
