# Peer review of "The Role of BRAF in Metastatic Colorectal Carcinoma–Past, Present, and Future"

_ijms, 2020, doi:10.3390/ijms21239001_

Round 1
Reviewer 1 Report
An exaustive and well-written review about BRAF mut CRC, regarding both preclinical and clinical aspects.
Just few considerations:
-regarding FOLFOXIRI+BV, the TRIBE2 trial did not confirm the previous results. Thus, it is not correct to consider this option as the standard. You should add this data.
-you should better explain results of subgroups analysis from VELOUR study.
Author Response
An exaustive and well-written review about BRAF mut CRC, regarding both preclinical and clinical aspects.
Just few considerations:
-regarding FOLFOXIRI+BV, the TRIBE2 trial did not confirm the previous results. Thus, it is not correct to consider this option as the standard. You should add this data.
-you should better explain results of subgroups analysis from VELOUR study.
Dear reviewer,
We really appreciate your constructive feeback which helps to increase the quality of the manuscript.
Accoriding to your suggestings we
- included in the revised manuscript the recently published data of the TRIBE2 trial and stated, that these results will certainly challenge the current guidelines. (page 7-8; line 303 – 307)
- elaborated the VELOUR results with regard to the analysed different mutation subgroups, highlighting the better response in BRAFmt vs BRAFwt compared to RASmt vs RASwt. (page 7; line 272– 278)
Reviewer 2 Report
The manuscript by Djanani et al. titled “The role of BRAF in metastatic colorectal carcinoma –past, present, and future” reports a detailed overview of current knowledge on the role of BRAF in metastatic colorectal carcinoma.
The authors firstly describe in details intracellular signalling cascades that can become drivers for tumor development such as RAS-RAF, PI3K-PKB and Wnt pathways. Then the current targeted therapies for CRC and the clinical relevance of molecular testing in CRC are described. The authors focused in particular on the prognostic and predictive value of BRAFV600E. Finally, they report BRAF targeted therapies, indicating the limits of the monotherapy approach and reporting recent data regarding combination therapies, such as combination of BRAF and MEK inhibition, combination of BRAF inhibitor dabrafenib and the anti-EGFR monoclonal antibody panitumumab, the combination of BRAF and EGFR and PI3K inhibition.
The literature is clearly reported and discussed.
The review is well written, complete and interesting for the readers of the Journal
Author Response
Dear reviewer,
we really appreciate your positive feedback.
kind regards
Manuel Maglione